# Optimization of an Open-Cell Foam-Based Ni-Mg-Al Catalyst for Enhanced CO$_2$ Hydrogenation to Methane

**Paulina Summa** [1,2], **Monika Motak** [2] and **Patrick Da Costa** [2,*]

1    Institut Jean Le Rond D'Alembert, Sorbonne Université, CNRS UMR 7190, 2 Place de la Gare de Ceinture, 78210 Saint-Cyr-L'Ecole, France; summa@fhi-berlin.mpg.de

2    Faculty of Energy and Fuels, AGH University of Science and Technology, Al. A. Mickiewicza 30, 30-059 Kraków, Poland; motakm@agh.edu.pl

*    Correspondence: patrick.da_costa@sorbonne-universite.fr

**Abstract:** In the presented work, the catalytic performance of a nickel catalyst, in CO$_2$ hydrogenation to methane, within a ZrO$_2$ open-cell foam (OCF)-based catalyst was studied. Two series of analogous samples were prepared and coated with 100–150 mg of a Mg-Al oxide interface to stabilize the formation of well-dispersed Ni crystallites, with 10–15 wt% of nickel as an active phase, based on 30 ppi foam or 45 ppi foam. The main factor influencing catalytic performance was the geometric parameters of the applied foams. The series of catalysts based on 30 ppi OCF showed CO$_2$ conversion in the range of 30–50% at 300 °C, while those based on 45 ppi OCF resulted in a significantly enhancement of the catalytic activity: 90–92% CO$_2$ conversion under the same experimental conditions. Calculations of the internal and external mass transfer limitations were performed. The observed difference in the catalytic activity was primarily related to the radial transport inside the pores, confirmed with the explicitly higher conversions.

**Keywords:** CO$_2$ methanation; nickel catalyst; open cell foams

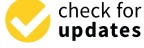



## 1. Introduction

The emission of greenhouse gases (GHG) is one of the biggest social concerns due to it being associated with climate change and an increase in the global mean surface air temperature (GSAT) [1]. During the last Conference of the Parties of the UNFCCC (COP 27) held in Egypt, it was decided to undertake rapid and sustained actions to reduce GHG emissions in order to limit the increase in GSAT to 1.5 °C compared to levels from the pre-industrial era [2].

One of the promising solutions for CO$_2$ utilization is its hydrogenation to methane (Equation (1)), where, from carbon dioxide and hydrogen, methane and water vapor are produced. Despite the exothermicity of the process, due to the kinetic limitations, the reaction is normally carried out in the temperature range between 250 and 450 °C [3].

$$4\,H_2 + CO_2 \leftrightarrow 2\,H_2O + CH_4 \qquad \Delta H^0_{298} = -165\ kJ/mol \qquad (1)$$

The increasing number of pilot plants and facilities for CO$_2$ methanation reaction brings confirmation of its utility as a perspective for the utilization of carbon dioxide. Some of the recent, large-scale projects are listed in Table 1. One can note that in the Audi plant in Werlte (Germany), the produced SNG is further applied as a fuel for vehicles [4].

**Table 1.** Current hydrogenation of $CO_2$ to $CH_4$ projects [5–8].

| Name | Location | Partners | Power Input | Status | Year |
|---|---|---|---|---|---|
| PtG test plant Stuttgart | Germany, Stuttgart | ZSW, IWES, Etogas | 250 kW | Pilot plant | 2012 |
| PTG test plant Rapperswil | Switzerland, Rapperswil | Erdgas Obersee AG, Etogas, HSR | 25 kW | Pilot plant | 2014 |
| E-gas/PtG BETA plant | Germany, Werlte | ZSW, Audi, EWE, IWES | 6300 kW | Commercial | 2013 |
| Copenhagen project of PtG | Denmark, Avedore Copenhagen | Electrochaea, BCH | 1000 kW | Commercial | 2016 |
| CO$_2$-SNG | Poland, Łaziska Górne | Tauron, Atmostat, CEA, Rafako | 4930 kW | Pilot plant | 2019 |
| Store and Go | Germany, Falkenhagen | Atmostat, Electrochaea, KIT, Uniper | 580 kW | Pilot plant | 2017 |

Considering the catalytic system suitable for $CO_2$ methanation reaction, the application of nickel brings the most optimal tradeoff between the obtained performance and its cost and availability. Additionally, nickel presents similar activity as that of the noble metals for this reaction, which makes it the optimal active center for a $CO_2$ hydrogenation reaction, presenting the ability to activate a strong C=O bond [9].

As reported elsewhere, a mixed oxide Ni–Mg–Al catalyst is very effective in $CO_2$ methanation, even without the further addition of a promoter. Such a type of uniform nano-oxides may be obtained throughout the thermal decomposition of hydrotalcites or, likewise, by the simultaneous solution combustion synthesis of all the precursors [10–12]. The coexistence of both aluminum oxide and magnesium oxide in the support phase is especially beneficial. Alumina is a source of the Lewis acidity necessary for the activation of oxygen in a carbon dioxide molecule (which has the nature of a Lewis base, being able to complete a pair of electrons) and the further intermediate steps of the reaction. Magnesium oxide, especially the Mg–O pair, is a source of medium-strength Lewis basic centers, on which bidentate carbonate is formed. Medium-strength basicity is also reported to be the one of the most influential factors enhancing $CO_2$ conversion [13–15].

Powder catalyst has been extensively studied as the most available solution for the $CO_2$ hydrogenation reaction; however, several constraints are associated with the pulverized bed, among which the most common one is a pressure drop over the reactor, especially if the fine size of particles is used. In a reaction such as $CO_2$ methanation, water is formed, and without the good permeability of the bed, its residence over the catalyst may destroy the structure, causing its deactivation [16,17]. Formation of hot spots and a lack of uniform heat dissipation from the catalytic bed lead to the overheating of some areas, resulting in an enhanced deactivation of the catalyst in those zones. Only a few studies dealt with the use of structured materials for $CO_2$ methanation [18–23]

Danaci et al. studied Ni/Al$_2$O$_3$ systems deposited on macroporous metal structures prepared with the robocasting technique—three-dimensional fiber deposition, forming different stack systems. The results showed that samples with the same stacking originating from a textural difference did not convey that to the changes in $CO_2$ conversion. In contrast, varying the stacking structure did improve the catalytic activity, due to the higher residence time distribution inside the channels. An additional factor enhancing the catalytic performance was the application of heat-conductive materials as the structured catalytic support [24]. Italiano et al., likewise, reported that improved thermal management resulting from the heat-conductive material of structured catalysts increases the activity in the $CO_2$ methanation reaction [25]. A particular focus to understand structure–heat transport relations inside an open-cell foam-based catalyst was reported by Sinn at al., who confirmed thermal conduction as the crucial heat-removal factor and, additionally, found that the strut diameter is the key parameter to improve heat transport and reduce possible hot spots' formation [26,27]. Fukuhara et al. investigated, among others, honeycomb-type Ni/CeO$_2$ prepared on an aluminum stacked-type-fin substrate with different cell density. It was observed that the stacked-type catalyst, based on fin with 200 cpsi (counts per square inches), was more active than that with 100 cpsi, confirming that the configuration of the honeycomb-fin and its cell density explicitly influences the catalytic performance [28]. Vita et al. deposited varying amounts of coating containing nickel on GDC (gadolinium-doped ceria) over a monolith. Samples covered with a low loading of the active phase

showed minor activity, explained by the insufficient layer of the catalyst. A higher loading of the coating led to an increase in the $CO_2$ conversion. Such a catalyst was characterized by a high surface-to-volume ratio, good interphase mass transfer, and low pressure drop [22]. Various configurations of metallic honeycomb-type stacked catalysts (plain-, stacked, segment-, and multi-stacked) were examined as a base for the $Ni/CeO_2$ catalyst. It was confirmed that the effects of the random flow channel and the gap distance were visibly enhancing the $CO_2$ conversion. Especially active was the multi-stacked system, in which moderate hot spots' formation was observed, eventually re-boosting the $CO_2$ conversion to a high level [21]. Series of honeycomb-based catalysts containing nickel supported on different binary oxides such as binary oxides $TiO_2$, $Al_2O_3$, $Y_2O_3$, and $CeO_2$ were tested toward $CO_2$ hydrogenation to methane. The most active catalyst contained 10 wt% of $Ni/CeO_2$, reaching 80% of $CO_2$ conversion at 298 °C [29]. A monolithic Ni foam catalyst coated with $Ni-Al_2O_3$ as a catalytically active phase was introduced by Li et al. This system obtained a remarkably high $CO_2$ conversion of 90% at 320 °C, and fluid dynamics calculation combined with experimental measurements confirmed explicit decrease in the hot spots' formation [30]. Commercial open-cell Ni foams with high pore density (75 ppi), further coated with $CeO_2$ via electro-precipitation and impregnated with Rh, were applied by Cimino et al. as a catalyst for $CO_2$ methanation. An important role was specially assigned to $CeO_2$, which facilitated the reduction of Rh and Ni and promoted the formation of oxygen vacancies. It was, likewise, crucial to enhance selectivity toward methane formation [31]. In the study presented by Ho et al., NiAl hydrotalcite-type materials containing La, Ce, and Y were coated by in situ electrodeposition on thermally conductive NiCrAl open-cell foams. The most active was the Ce-promoted catalyst, obtaining improved catalytic performance in the low-temperature region. In general, promoters improved surface basicity, Ni dispersion, and reducibility. Open-cell foam-based samples were compared with a powder catalyst, resulting in similar physicochemical properties. Additionally, the catalytic result of the structured samples was compared to that of the analogous pellet samples, resulting in improved catalytic activity compared to the latter, which could be assigned to an optimized heat transfer [32]. Among the other reported studies on the OCF-based catalyst for $CO_2$ hydrogenation, Frey et al. investigated the $Ni-Ru/CeO_2-ZrO_2$ catalyst based on SiC and $Al_2O_3$ foams, Balzarotti et al. investigated $RhNi/CeO_2$ systems, and Italiano et al. investigated the $Ni/CeO_2-ZrO_2$ catalyst [25,33–35].

In some sustainable applications, open-cell foam as a structured catalyst is considered to be an interesting alternative to the monolith or honeycomb structures [36]. In fact, because of the chaotic, random configuration of the pores, both axial and radial dispersions of heat and mass are favored. The distributions of temperature and reactants are better, leading to faster kinetics and, eventually, higher catalytic activity [37]. Additionally, as expected for structured catalysts, open-cell foams allow for a reduction in pressure drop. Commercial OCFs are used in the metallurgical industry as ceramic filters for steel, iron, and non-ferrous alloys, characterized by extreme heat resistance [38]. It was already widely reported in the literature that an open-cell foam-based catalyst is extremely successful in the process of the catalytic combustion of methane [39–41], reforming biogas [42,43]. Additional interesting applications can be the steam reforming of methane (SMR) [44–48] and the Fischer–Tropsch synthesis (FTS) [49–53]. For the former reaction, similar catalytic systems to those presented in this article were investigated. Cristiani et al. studied 10% nickel on $MgAl_2O_4$ oxide coated on FeCrAlloy foams. The obtained washcoat layers were rather homogeneous and well-adherent to the metallic foam. Additionally, the system was active in the steam reforming of the methane reaction, approaching thermodynamic equilibrium above 450 °C with high space velocity [44]. Balzarotti et al. compared different open-cell foam metallic structures such as FeCrAlY and copper open-cell foams as a support for the SMR rhodium catalyst. It was observed that copper matrices enabled higher conversions and reduced temperature gradients, allowing for the more flexible operation of the reformer with improved thermal efficiency [45,47].

Regarding the Fischer–Tropsch reaction, Aguirre et al. intensively focused on the determination of mass transport limitations in the open-cell foam-based bed, using $Co/Al_2O_3$ and $Co/TiO_2$ as the catalytically active coating. The studied samples were active in FTS, showing performance typical for Co-based systems. However, with an increment of the coating thickness, the reaction rate increased, together with the observed deviation in regard to the selectivity toward methane and loss of the catalyst's efficiency. It was associated with strong diffusion limitations, while the influence of the external mass transfer and the heat transfer were negligible under the studied conditions [50]. Fratalocchi et al.'s work mainly focused on the heat transfer inside the open cellular/foam bed in the FTS reaction, due to its strongly exothermic nature and the need for temperature control. The studied $Co/Pt/Al_2O_3$ system, in the form of microspheres or pellets, was packed with a foam or open cellular structure loaded in the tubular reactor. The obtained results confirmed that the use of foam fabricated with a thermally conductive material allows an enhanced heat transfer performance and controls the mean temperature inside the catalytic bed [52,53].

The presented work deals with the influence of the application of $ZrO_2$ open-cell foams, as a carrier for the mixed oxide Ni–Mg–Al catalyst. $ZrO_2$ foams were chosen due to the relatively high mechanical strength, especially in tetragonal form [54], which facilitated the preparation of the reactor for the catalytic tests, in which foam was directly closed in the gas tube blown around it. The deposited coating containing Ni–Mg–Al mixed oxides was generally reported as an extremely active and suitable material for $CO_2$ hydrogenation to methane [10,11].

## 2. Results

### 2.1. Characterization of the Open-Cell Foam-Based Catalysts

The XRD diffractogram for the bare open-cell foam is presented in Figure S1. Due to the exceptional hardness of $ZrO_2$ foams, before XRD they were crushed in an agate mortar and later pressed to form pellets, to obtain a geometry adequate for measurement. However, the pellets were not totally homogeneous, containing larger pieces of an extremely rigid foam structure.

All of the presented reflections correspond to the general structure of zirconium oxide; however, two types of crystalline systems were distinguished—monoclinic and cubic zirconium oxide. Monoclinic $ZrO_2$ (ICDD 01-078-0047) was represented by major reflections at 2θ with corresponding (001) planes at 17.43 deg (100), 24.07 deg (011), 24.46 deg (110), 28.19 deg (−111), 31.47 deg (111), 34.15 deg (002), 34.45 deg (020), 35.28 deg (200), and 50.13 deg (220). Cubic $ZrO_2$ (ICDD 00-049-1642) was assigned to the reflections at 2θ with (001) planes at 30.12 deg (111), 34.96 deg (200), 50.22 deg (220), 59.74 deg (311), and 62.68 deg (222). The registered diffractogram was characterized by tall and narrow peaks, suggesting large crystallites. No scattering characteristic for the amorphous phase was recorded.

XRD diffractograms of OCFs with the deposited Ni–Mg–Al nickel oxides are reported in Figure 1. Most of the registered peaks were assigned to cubic and monoclinic zirconia, as discussed above. In two subsequent steps of synthesis, Mg–Al mixed oxides and NiO were deposited on the bare foams. Magnesium and aluminum were expected to form a $Mg(Al)O_x$ oxide with a periclase-like structure that is usually identified at similar 2θ values as NiO. The characteristic for MgO reflections (ICDD 00-045-0946) with corresponding (001) planes was at 36.9 deg (111), 42.9 deg (200), 62.3 deg (220), 74.7 deg (311), and 78.6 deg (222), while for NiO it was 37.2 deg (111), 43.4 deg (200), 62.9 deg (220), 75.4 deg (311), and 79.4 deg (222). The registered peaks were shifted in the direction of NiO, confirming its domination in the phase deposited on top.

Low-temperature $N_2$ sorption was performed on bare $ZrO_2$ foam with 45 ppi and 15_150_45 and 15_150_30 catalysts. Due to the very low specific surface area, it was not possible to draw a $N_2$ adsorption–desorption isotherm for a bare $ZrO_2$ foam due to the limitations of the apparatus and method used. Therefore, the specific surface area of the started material ($ZrO_2$ foam) is assumed to be less than 1 $m^2/g$. The 15_150_45 and

15_150_30 catalysts were considered as the representative to perform the textural study on the foams covered by the active phase. The obtained adsorption–desorption isotherm (Figure 2) was classified as IV type according to IUPAC, which was confirmed by the presence of small hysteresis loop, suggesting the formation of mesopores [55]. The specific surface area for the catalyst based on the 45 ppi foam was ca. 14 m$^2$/g, while for that based on the 30 ppi foam it was ca. 9 m$^2$/g. Since the geometric surface area of the foams used is negligible, additionally the specific surface area per gram of coating was calculated. In this case, the surface area of the deposited oxide phase was found to be ca. 165 m$^2$/g$_{coating}$ for the 15_150_45 catalyst and ca. 111 m$^2$/g$_{coating}$ for the 15_150_30 catalyst. These values are comparable with the specific surface area of the similar powder Ni–Mg–Al mixed oxide catalyst prepared via solution combustion synthesis, as reported elsewhere [11]. The total pore volume and an average pore diameter are listed in Table 2, as orientational values, considering that such a low specific area is ladened with a large measurement error (generally considered to be 10%).

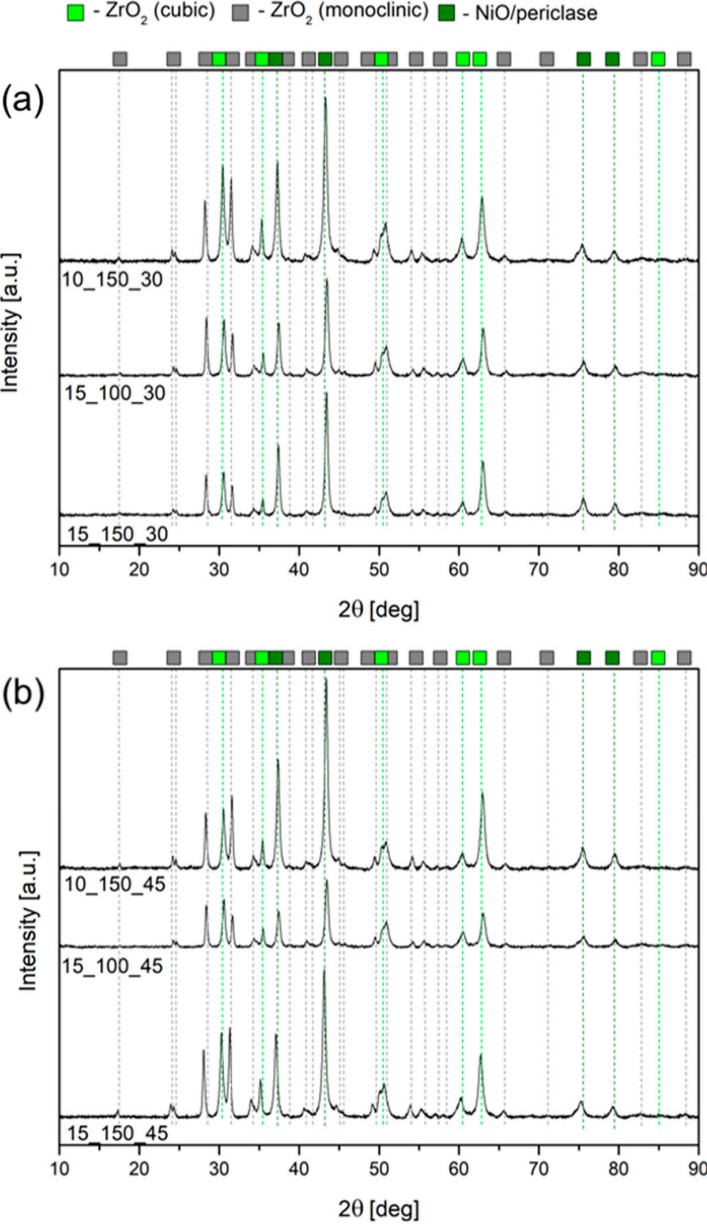

**Figure 1.** XRD diffractograms for open-cell foams (**a**) with porosity of 30 ppi and (**b**) with porosity of 45 ppi, after deposition of Ni–Mg–Al mixed oxides.

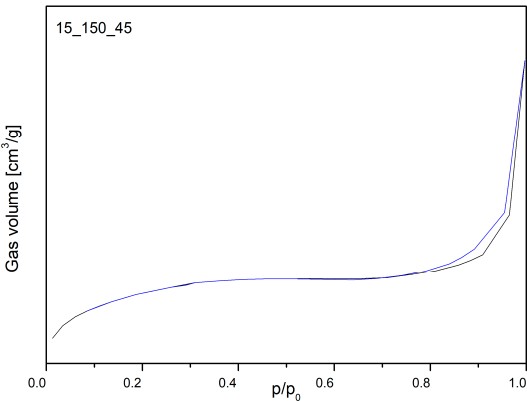

**Figure 2.** Low-temperature $N_2$ sorption isotherm for 15_150_45 catalyst.

**Table 2.** Specific surface area and porous properties of bare OCF and the studied catalysts.

| Catalyst | $S_{BET}$ (m²/g) | $S_{BET}$ (m²/g$_{coating}$) | Total Pore Volume (cm³/g) | $d_p$ (nm) | $V_{mesopores}$ (cm³/g) |
|---|---|---|---|---|---|
| $ZrO_2$ foam | <1 | | - | - | - |
| 15_150_45 | 14 | 165 | 0.016 | 4.6 | 0.010 |
| 15_150_30 | 9 | 111 | 0.010 | 4.7 | 0.006 |

The $H_2$ temperature-programmed reduction (TPR) profiles of OCF-based catalysts are reported in Figure 3. $ZrO_2$ foam per se was, likewise, studied for $H_2$-TPR, although none of the reducible phases were registered. For the catalysts coated with mixed oxides, all the studied samples showed only one broad and complex peak in the temperature range from 300 to 450 °C, corresponding to the reduction of $Ni^{2+}$ to $Ni^0$ from nickel oxide. The lack of peaks reducible above 500 °C suggests a lack of the dissociation of nickel to the supporting $Mg(Al)O_x$ periclase-like structure.

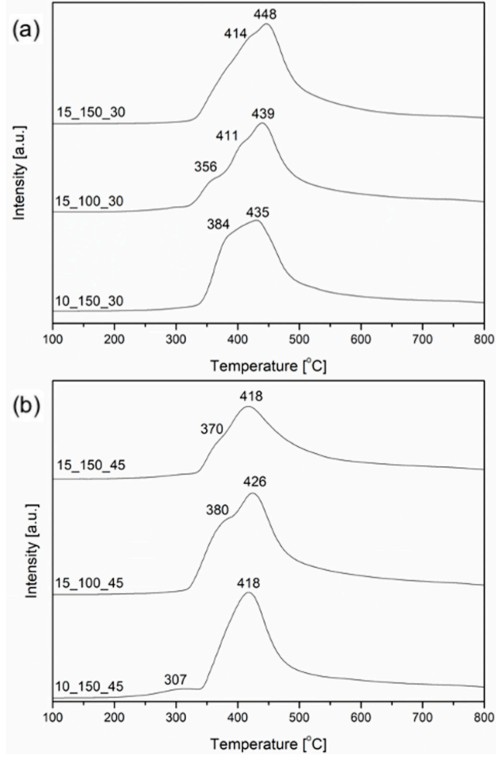

**Figure 3.** $H_2$-TPR profiles for open-cell foam-based catalysts with porosity of (**a**) 30 ppi and (**b**) 45 ppi.

Samples deposited onto 30 ppi OCFs showed reduction peaks at somewhat higher temperatures than the corresponding materials prepared with 45 ppi OCFs. They may eventually be associated with the slightly lower specific surface area of the former. Considering the overall thickness of the deposited coating, mass transfer limitations should not affect reducibility.

It was observed that the reduction of $Ni^{2+}$ to $Ni^0$ can take place in two or three correlated steps. For the 15_150_30, 10_150_30, 15_150_45, and 15_100_45 catalysts, two subsequent steps were recognized; the first one may correspond to the reduction of NiO agglomerates characterized by weaker interaction with the support, while the second one suggests that the reduction of isolated nickel oxide crystallites was strongly interacting with the support [56]. In the case of the 15_100_30 sample, a three-step reduction was registered, where the second and third reaction stages can be explained as those of the former samples. However, the first one, corresponding to the low-temperature shoulder with the maximum at 356 °C, may be linked with the reduction of weakly bonded NiO crystallites. For the sample denoted 10_150_45, a small shoulder with a reduction temperature maximum at 307 °C was registered. The latter peak probably corresponds to the reduction of well-dispersed and weakly bonded NiO and was followed by a large peak with its center at 418 °C, typical for the reduction of bulk nickel oxide [56].

X-ray diffractograms (XRD) for reduced OCF-based catalysts are presented in Figure 4. Monoclinic $ZrO_2$ (ICDD 01-078-0047) and cubic $ZrO_2$ (ICDD 00-049-1642) were confirmed with the same reflections as shown in Figure S1. In addition, two phases confirming the presence of coating were recognized—periclase-like oxide (ICDD 00-045-0946) and metallic nickel (ICDD 03-065-0380). For the reduced catalysts, reflections located at 36.9 deg (111), 42.9 deg (200), 62.3 deg (220), 74.7 deg (311), and 78.6 deg (222) were typical for MgO and did not show any shift in the direction of nickel oxide, confirming the reduction of the majority of nickel oxide species.

The presence of the $Ni^0$ phase was endorsed by the three sharp reflections located at the 2θ of 44.3 deg (111), 51.7 deg (200), and 76.1 deg (220). The average size of the nickel crystallites was calculated using Scherrer's formula and is presented in Table 3. The average $Ni^0$ crystallite size for all samples is in the same range, from 7 to 11 nm. In comparison to the solution-combustion-synthesis-derived powder catalysts reported elsewhere [11], the obtained crystallite size is remarkably smaller. This may be associated with the significantly lower reduction temperature that was applied (500 °C).

**Table 3.** Distribution of basic sites for reduced open-cell foam-based catalysts and $Ni^0$ crystallite based on Scherrer's formula (XRD).

| Sample | Weak (μmol/g) | Medium (μmol/g) | Strong (μmol/g) | Total (μmol/g) | $Ni^0$ Crystallite Size (nm) |
|---|---|---|---|---|---|
| 15_100_30 | 1 | 6 | 5 | 13 | 8 |
| 10_150_30 | 15 | 9 | 23 | 47 | 11 |
| 15_150_30 | 7 | 7 | 6 | 20 | 9 |
| 15_100_45 | 9 | 10 | 9 | 28 | 10 |
| 10_150_45 | 7 | 7 | 20 | 35 | 8 |
| 15_150_45 | 18 | 21 | 5 | 44 | 7 |

The $CO_2$ temperature-programmed desorption (TPD) profiles for the studied catalysts are presented in Figure 5. The profiles were deconvoluted into three Gaussian curves associated with the weak, medium strength, and strong basic sites, with the maxima of desorption in the temperature range of 136–178 °C, 202–283 °C, and 314–385 °C, respectively. The desorption range for each type of basic center was remarkably vast, in the case of the examined series of catalysts. In particular, a high desorption temperature for weak sites was registered for the 10_150_30, 15_150_30, and 15_150_45 catalysts, for which the coating was characterized by the thickest layer of Mg–Al mixed oxides. This could have led to the broad interaction between $CO_2$ and the available sites on the surface. For these

samples, the desorption temperature of $CO_2$ from medium-strength sites was also very high. The remaining samples, 15_100_30, 15_100_45, and 10_150_45, were characterized by a thinner layer of the deposited oxide coating and resulted in maximum $CO_2$ desorption at lesser temperatures. The desorption temperature range, however, recorded for the OCF-based samples was similar to that of Ni–Mg–Al hydrotalcite-derived catalysts, as reported elsewhere [10,57].

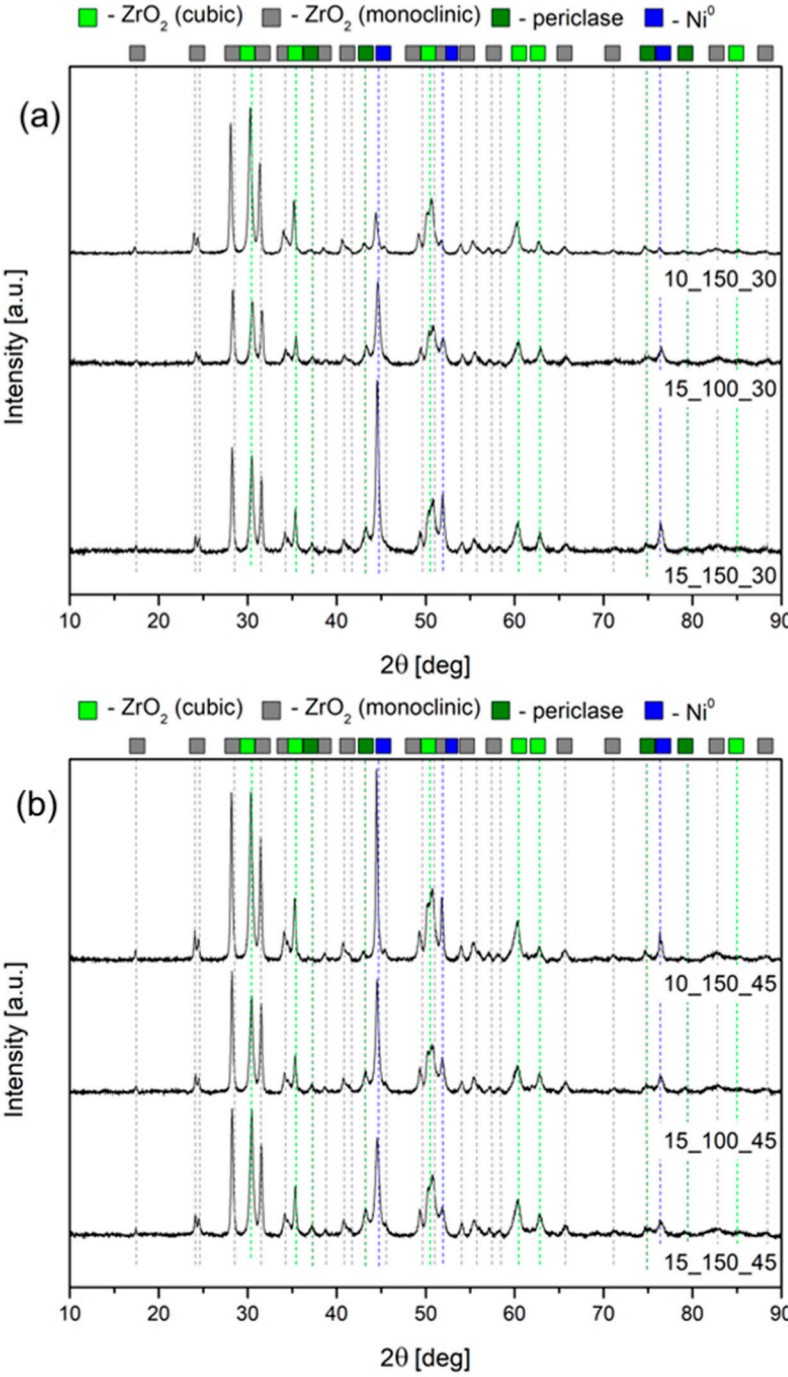

**Figure 4.** XRD diffractograms for reduced catalysts based on open-cell foams (**a**) with porosity of 30 ppi and (**b**) 45 ppi.

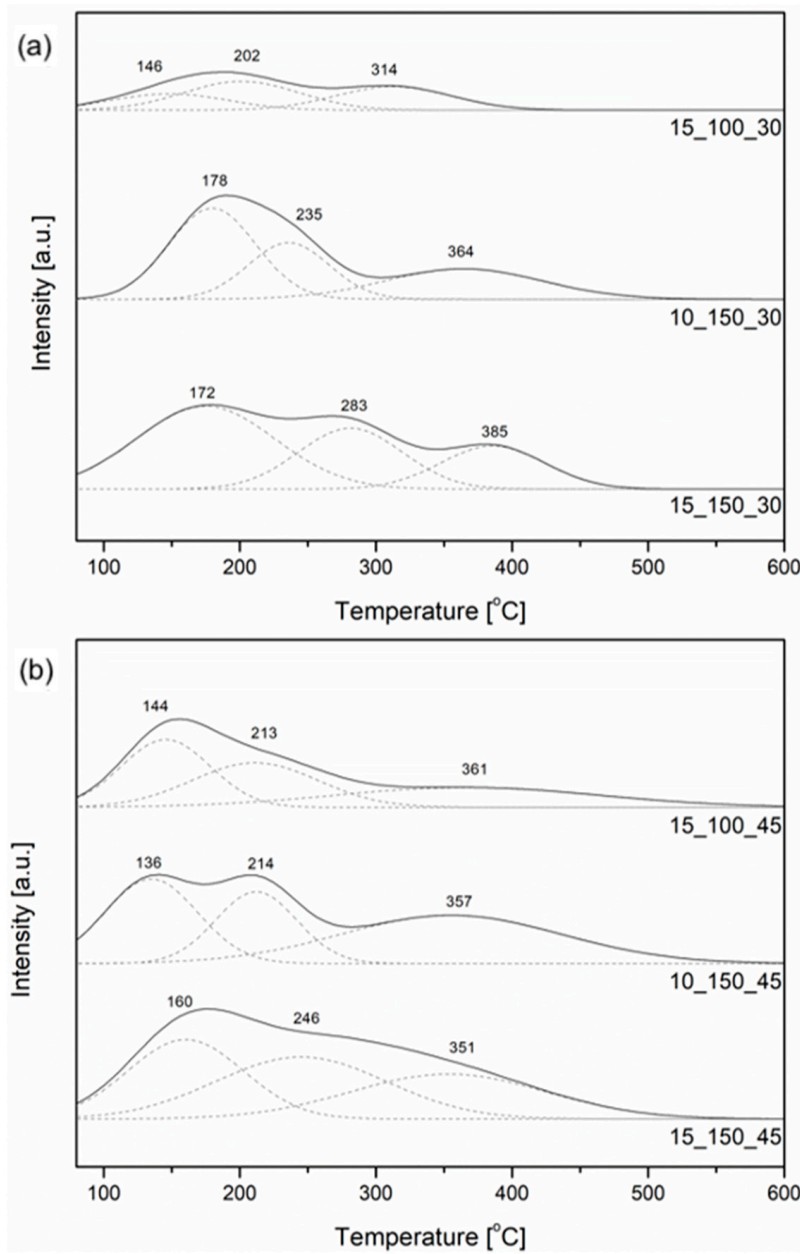

**Figure 5.** $CO_2$-TPD profiles for the reduced catalysts based on open-cell foams (**a**) with porosity of 30 ppi and (**b**) 45 ppi.

In Table 3, the contribution of each type of basic site is listed. The studied catalysts did not show an extensive number of basic sites per unit of mass. Predominantly, a higher number of basic sites was registered for catalysts based on OCFs with porosity of 45 ppi. The 15_100_30 catalyst showed the lowest number of basic sites, probably both due to the low loading of the Mg–Al mixed-oxide phase and the smaller exposed surface (due to the use of 30 ppi foam). The 15_150_30 and 15_100_45 samples resulted in a comparable contribution to each type of basic center on the surface. Moreover, the 10_150_45 catalyst was dominated by strong basic sites. It is worth noting that, in this sample, the Mg–Al mixed-oxide layer may be expected to present higher surface availability as a result of its larger deposited amount, the higher surface area of the foam, and, eventually, the lower loading of nickel. An analogous effect was observed in the 10_150_30 sample, which, additionally, showed a relatively large content for its weak basic sites. Finally, the 15_150_45 catalyst was dominated by medium-strength and weak basic centers.

The SEM images of the reduced open-cell foam-based catalysts are compared in Figure 6. The images Figure 6(1a),(2a) and registered at low magnification clearly show the cells of the foams coated with an active phase. It may be observed that the 15_150_30 catalyst shows a more consistent coating with a uniform thickness. In contrast, at the given scale, it is also visible that the active phase of the 15_150_45 catalyst is less ordered and smooth, with cracks and spots where the washcoat is cumulated. This result can be justified by the specificity of the synthesis—foams with 45 ppi show an explicitly smaller pore diameter. During the formation of the coating containing the catalytically active phase, the solution with precursors was probably blocked in certain places where the liquid could not leave the structure (due to the surface tension) or spontaneously evaporate and formed noticeable cracks and lumps. Other factors include the relatively massive layer of deposited oxides and the multiplicity of the reproduced steps of coating deposition. To conclude, open-cell foams with porosity of 30 ppi demonstrated remarkably larger cells, and, presumably due to this property, the solution was able to evenly cover the surface.

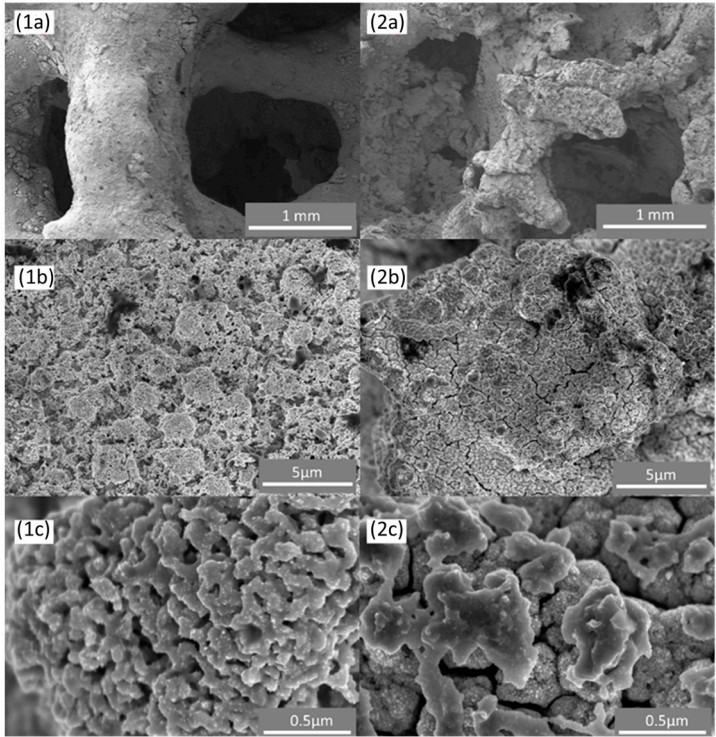

**Figure 6.** SEM images of the reduced open-cell foam-based catalysts (**1a–1c**) 15_150_30 and (**2a–2c**) 15_150_45 ppi.

Furthermore, images taken at a high magnification show the differences between the active phases formed on the surface of foam with 30 and 45 ppi. Thus, it is visible that the washcoat on the 30 ppi open-cell foam is more uniform and less condensed than the one on the 45 ppi foam.

### 2.2. Catalytic Tests

The results of the $CO_2$ methanation tests performed on the series of catalysts deposited on foams with 30 ppi are compared in Figure 7. At 250 °C, none of the catalysts showed activity expressed as $CO_2$ conversion. At 300 °C, the lowest conversion, of 33%, was observed for the 15_100_30 catalyst, while the highest conversion, of 49%, was observed for the 15_150_45 catalyst. The activity of 10_150_45 was similar to that of the latter. For temperatures higher than 350 °C, all the catalysts led to $CO_2$ conversion close to the values of thermodynamic equilibrium (Table S1).

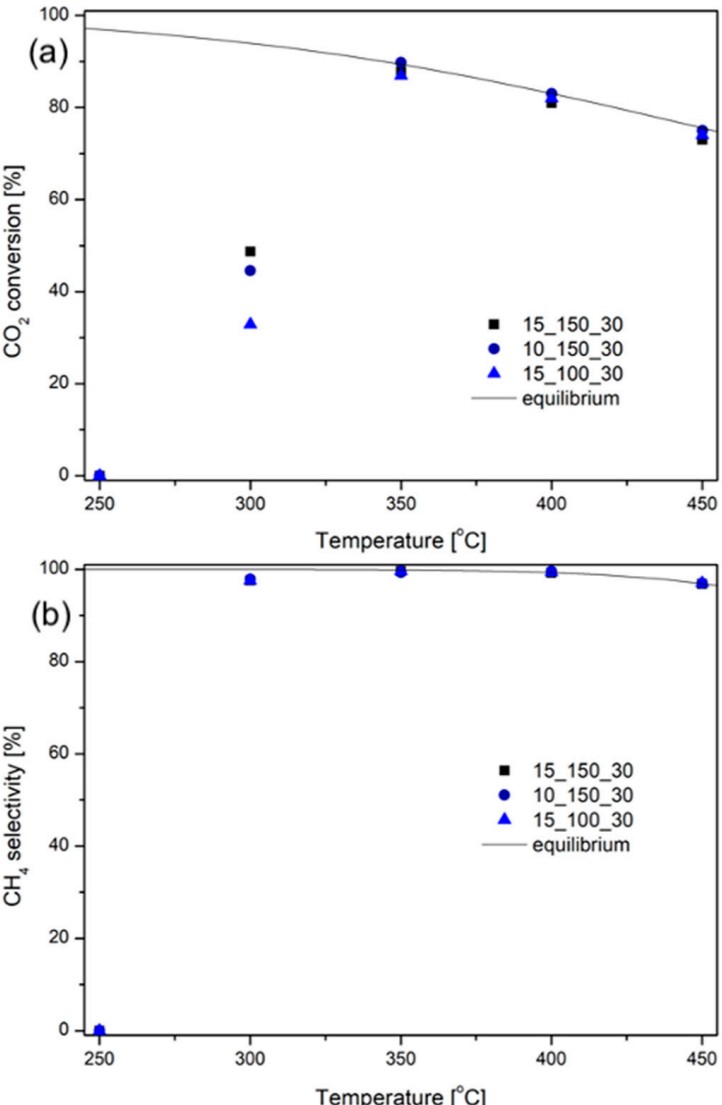

**Figure 7.** (**a**) $CO_2$ conversion and (**b**) $CH_4$ selectivity of catalysts based on 30 ppi OCFs.

It should be noted that, at the initial test temperature (250 °C), none of the samples showed methane formation due to the lack of converted $CO_2$. Such low activity at 250 °C could be related to kinetic limitations.

The selectivity to methane of the tested series of catalysts based on the 30 ppi OCFs at 300 °C was very high (ca. 98%). At 350 °C, all samples reached thermodynamic equilibrium, in the case of selectivity toward methane, and kept the maximum level of methane formation.

Figure 8 shows the results of $CO_2$ methanation tests for catalysts based on the 45 ppi open-cell foams. At 250 °C, no activity was registered, similarly to those based on OCFs with porosity of 30 ppi. A pronounced increase in $CO_2$ conversion was registered at 300 °C, up to 92% for 15_150_45 and 10_150_45. Above that temperature, the activity obtained for all the catalysts was comparable and close to the values of thermodynamic equilibrium. Complementary observations regarding the selectivity to $CH_4$ can be made. At 250 °C, no methane formation was observed in the aftermath of the lack of $CO_2$ conversion. At 300 °C and above, all the catalysts obtained equilibrium values of selectivity toward $CH_4$.

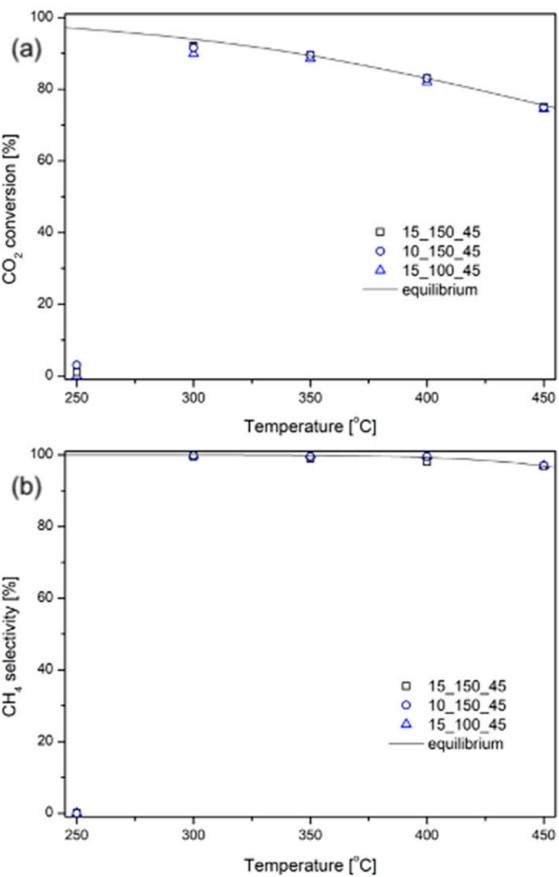

**Figure 8.** (**a**) $CO_2$ conversion and (**b**) $CH_4$ selectivity of catalysts based on 45 ppi OCFs.

Stability tests were performed (5 h at 300 °C) on the 15_150_30, 15_150_45, and 15_100_45 samples. The results are presented in Figure 9. The values obtained for $CO_2$ conversion and methane selectivity are similar to those from temperature-programmed surface reaction (TPSR) tests.

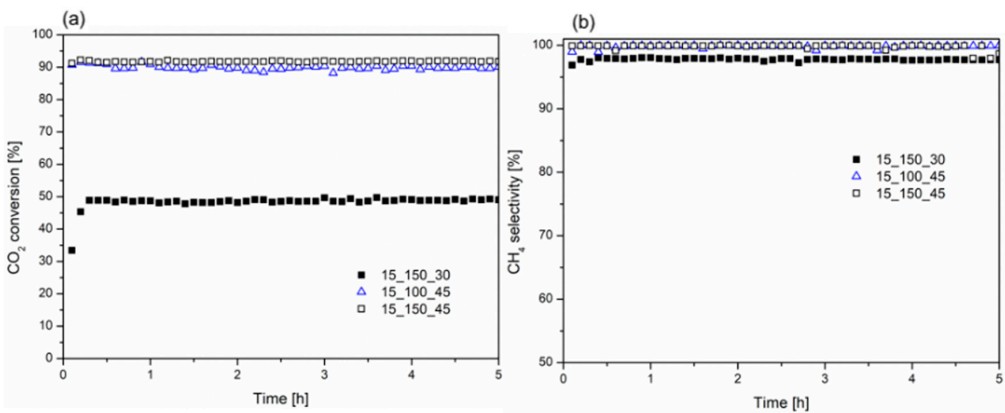

**Figure 9.** Five hour stability tests performed at 300 °C on OCF-based catalysts.(**a**) $CO_2$ conversion and (**b**) $CH_4$ selectivity.

Both parameters remained stable during the time on stream for all of the investigated catalysts, which suggests lack of deactivation under the studied test conditions during at least 5 h of time on stream.

### 2.3. Post-Run Characterization

The XRD diffractograms for the post-run catalysts are compared in Figure 10. As for the above-mentioned reduced catalysts, four phases were recognized, such as monoclinic $ZrO_2$ (ICDD 01-078-0047), cubic $ZrO_2$ (ICDD 00-049-1642), periclase-like oxide (ICDD 00-045-0946), and metallic nickel (ICDD 03-065-0380). No exemplary scattering for the amorphous phase and no typical reflections for carbon deposits characterized with crystalline structure were registered. The reflections, sharp and symmetric, assigned to both types of zirconia were unchanged, confirming large crystallites and a high level of crystallinity for the foam, suggesting that it does not undergo any structural changes during the reaction. Peaks related to the presence of the periclase-like oxide were short, symmetrical, and wide, suggesting a small crystallite size. The most intense reflections were correlated to metallic nickel. The average $Ni^0$ crystallite size, calculated with Scherrer's equation, is presented in Figure 10. On all of the post-run catalysts based on the comparable size of 45 ppi foam nickel particles, ca. 9–10 nm was observed. Thus, only a slight increase in particle size was registered, compared to the reduced samples. Similar conclusions can be drawn for the 15_100_30 and 10_150_30 samples. The catalyst that displayed a certain increment in the size of nickel crystallites was 15_150_30 (from 9 nm on the reduced catalyst to 15 nm on the post-run), which suggests, in such a case, the explicit sintering of $Ni^0$, probably due to the remarkably large surface density of nickel, taking into account the reduced exposed surface of 30 ppi foam.

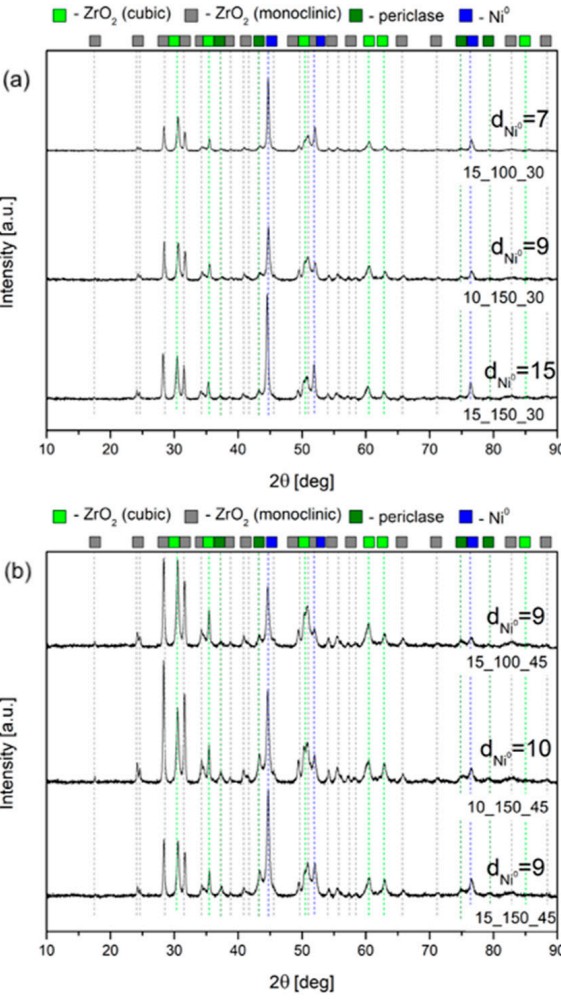

**Figure 10.** XRD diffractograms for post-run catalysts based on open-cell foams (**a**) with porosity of 30 ppi and (**b**) 45 ppi.

### 3. Discussion

As mentioned before, the hydrogenation of $CO_2$ to methane is a kinetic limited reaction, occurring only at the interface between the gas phase and the surface of the catalyst. Possibly, the reduced size of the nickel crystallites and the availability of the basic sites are enhancing the surface reaction for the chosen systems. In contrast, in the case of open-cell foam-based catalysts, an increment in the thickness of the deposited coating may result in the internal mass transfer rate being higher than that of the studied hydrogenation reaction. To minimize gas transport limitations, the thickness of the deposited catalytically active phase should be optimized, since it does influence the geometrical properties of the synthesized open-cell foam-based catalyst [58].

The open-cell foam-based $CO_2$ hydrogenation system is expected to operate under a laminar flow, characterized by a Reynolds number below 400; therefore, carbon dioxide mass transfer in the gas phase is principally dependent on molecular diffusion [59,60].

During $CO_2$ methanation, the feed gas mixture flows through the structure of the foam, while the reactants are expected to diffuse to the surface of the mixed oxide coating from the feed. In order to prevent the formation of a concentration gradient between the bulk gases and the reactive surface, the aforementioned transport needs to be faster than the surface reaction. In an extreme case, the distribution of reactants within the cell of the foam is not homogenous, which, in consequence, leads to a decrease in the $CO_2$ conversion. A mass transfer amid bulk feed gases and the coating may be enhanced by reducing the hydraulic diameter or promoting molecular diffusion [42].

In comparing the catalytic tests results performed on the OCF-based catalyst, as presented in Figures 6 and 7, one can conclude that a possible mass transfer limitation is present. The 30 ppi foam-based catalysts resulted in significantly lower activity in the $CO_2$ methanation reaction at 300 °C than those based on the 45 ppi foam, especially when taking into account the fact that the number of active phases was comparable.

The mass transfer inside the catalyst was analyzed by the dimensionless numbers and analysis involving the characteristic time, according to the work of Italiano et al. [42,61].

The foam parameters and methodology applied for the estimation of internal and external mass transfer limitations are presented in Section S2.

In Figure S3, the open-cell foam with porosity of 30 ppi is presented, with the main geometric parameters used for further calculations. The open-cell foams' structure consists of cell units, branching off in all space dimensions, limited by struts. The average pore area and strut thickness were calculated based on images of bare and coated foams. The pores were assumed to present a circular shape. The kinetics on which the calculations were based assumed a first-order reaction.

Damkȏhler number I (Da-I) is considered as a parameter evaluating the relations among the kinetics of the reaction and the reactor design, in particular the length, and the operating conditions. In Figure 11, the values of Da-I for the 15_150_30 and 15_150_45 catalysts are compared at the test temperatures. The temperature of 250 °C was excluded due to the explicit lack of activity registered for both catalysts. Similarly, 400 and 450 °C temperatures were excluded due to the conversion approaching thermodynamic equilibrium. A Da-I number above 1 implies that the reactants within the feed have had enough time to complete the catalytic reaction. A Da-I number lower than 1 suggests a deficit of contact time between the foam and feed to undergo a successful conversion. In the case of both 30 ppi foam and 45 ppi foam, the value of Da-I is visibly higher than 1, suggesting sufficient time for the reactants to complete the reaction, due to the relatively large length of the applied foam.

Consequently, based on Damkȏhler number II (Da-II), the pore dimensions necessary for the effective use of the reactor are estimated. Da-II values below 0.1 imply a negligible effect of the external mass transfer limitations, considered to be less than 3%. Furthermore, the low values of the Da-II number are a confirmation of the rapid radial transport of the reactants from the bulk toward the catalytically active surface, suggesting the uniform distribution of the reactants. The values of the Da-II number for the studied 15_150_30

and 15_150_45 samples are compared in Figure 12. It may be noted that the catalyst characterized by the larger pores of the applied foam, 15_150_30, showed Da-II values above 0.1 at 300 °C. This may be because the distribution of the reactants in the chosen microreactor was not uniform, and the time necessary for the radial transport of reactants to the active phase was not proportionate at that temperature. In contrast, the 15_150_45 catalyst, characterized by significantly smaller pores, resulted in Da-II values below 0.1. In the latter case, due to the significantly smaller dimensions of the pores, resulting in the shortened distance between the bulk and surface of the catalyst, radial transport is sufficient. At 350 °C, both catalysts resulted in Da-II numbers below 0.1.

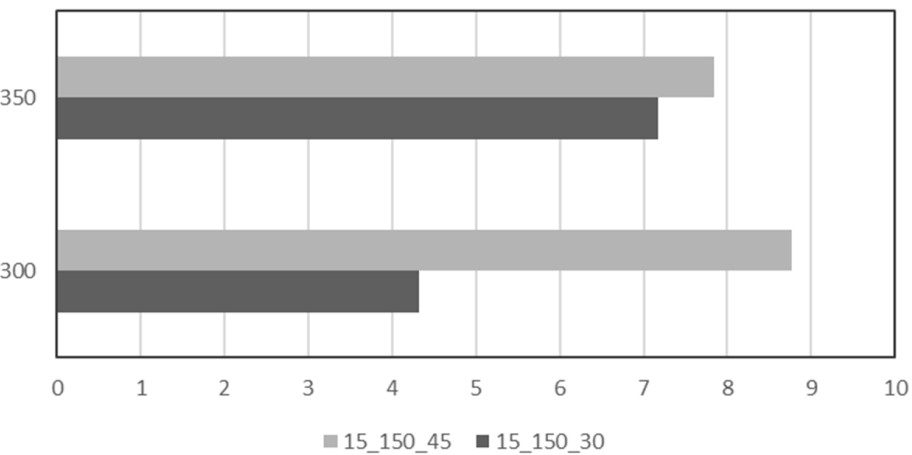

**Figure 11.** Values of Da-I number for 15_150_30 and 15_150_45 catalysts.

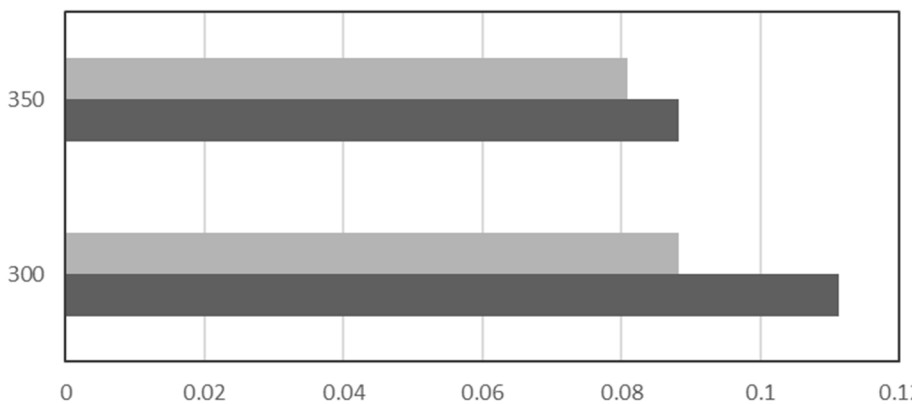

**Figure 12.** Values of Da-II number for 15_150_30 (dark grey) and 15_150_45 catalysts (light grey).

Moreover, as a confirmation of the Da-II number, the Carberry criterion was introduced. Its desired values are below 0.05. Similarly to Damköhler number II, the values of the Carberry number (Figure 13) for the catalyst deposited on 30 ppi foam are above the limit of 0.05 at 300 °C, suggesting radial mass transfer limitations. In contrast, for a sample based on 45 ppi foam, results below 0.05 suggest negligible external mass transfer limitations, due to the significantly smaller dimension of the foam pores. At 350 °C, both samples resulted in a Ca number below 0.05, which is in agreement with the Da-II number values presented above.

Damköhler number III (Da-III) informs about the diffusion limitations in the coating layer (Figure 14). Values lower than 1 allow for excluding the internal diffusion limitations under the studied conditions. The Da-III number for both catalysts is below 1, suggesting minor diffusion in the deposited active phase of the catalyst. The Weisz–Prater criterion was applied as an evaluation of the Da-III number, and the results are presented in Figure 15. As for the former dimensionless number, values below 1 indicate the acceptable thickness of the coating, minimizing internal mass transfer limitations. Likewise, both of the compared

catalysts resulted in values clearly below a boundary level of 1, allowing the assumption that the deposited active phase had the optimal thickness.

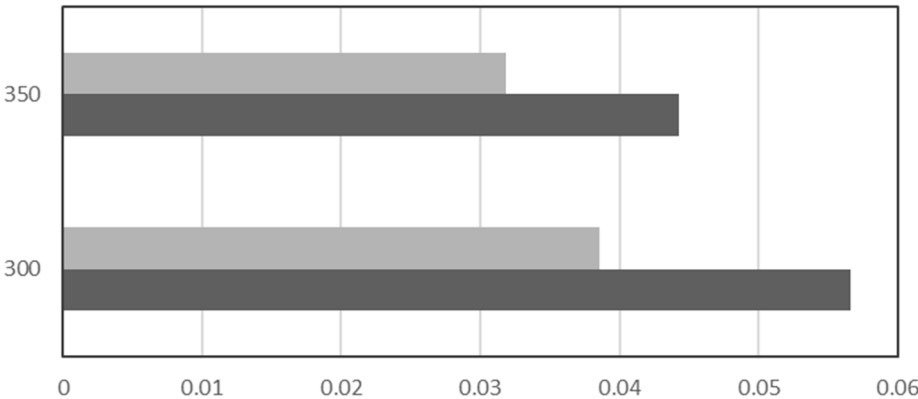

**Figure 13.** Values of Carberry number for 15_150_30 (dark grey) and 15_150_45 catalysts (light grey).

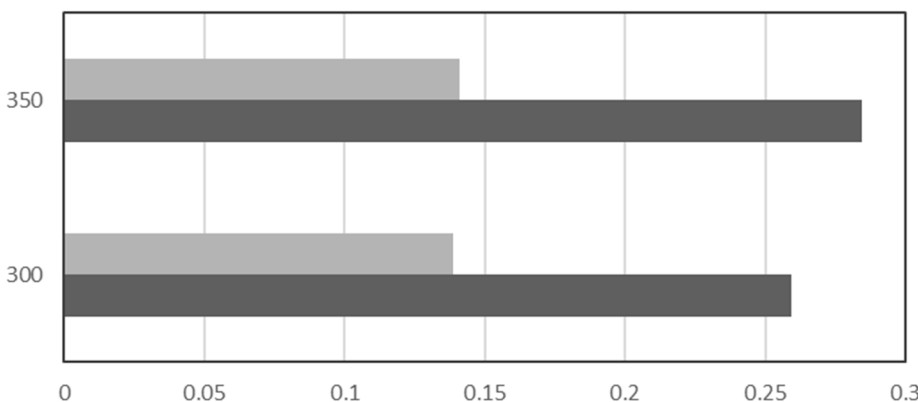

**Figure 14.** Values of Da-III number for 15_150_30 (dark grey) and 15_150_45 catalysts (light grey).

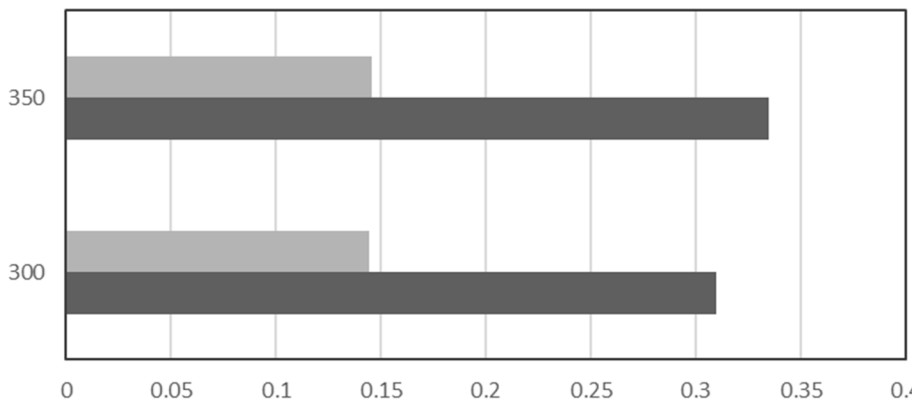

**Figure 15.** Values of Weisz–Prater criterion for 15_150_30 (dark grey) and 15_150_45 catalysts (light grey).

As reported elsewhere, for the Ni–Mg–Al mixed oxide system, such physicochemical properties as specific surface area and nickel dispersion are less influential factors regarding the $CO_2$ conversion of the methanation catalyst than, i.e., surface basicity [11,12]. Considering the differences of basic sites' distribution throughout the series of samples based on foams with 30 and 45 ppi, it is apparent that despite the varying availability of the surface basic centers associated with the different coating thickness, catalysts based on the same geometry of foam show comparable catalytic activity. It was strongly pointed out in the literature that the changes in the geometry of the structured catalyst are visibly affecting the activity, associating it with the diffusion of gases through the catalytic bed and eventual

heat transfer [24–28]. Thus, the presented conclusions are implying that the geometric dimensions, which are the most significant differentiating factor, are affecting the catalytic activity of the structured catalyst.

## 4. Materials and Methods

### 4.1. Synthesis of the Catalyst

Commercial $ZrO_2$ ceramic open-cell foams with length of 30 mm and diameter of 9 mm were supplied by Lanik s.r.o. (Boskovice, Czech Republic) with two types of porosity, 30 and 45 pores per inch (ppi). Prior to the deposition of the catalytically active phase, ceramic foams were purified in the sonic bath, using the solution of acetone in water (50% vol./50% vol.), and later dried at 80 °C for 6 h in static air. The coating was deposited in two subsequent steps. Firstly, the uniform layer of Mg–Al mixed oxides was produced on the foams with solution combustion synthesis, followed by calcination for 5 h at 500 °C in static air. Secondly, the layer of nickel oxide was deposited all over the foam coated with Mg–Al mixed oxides, also with solution combustion synthesis. For that purpose, 20 cm³ of 2 M solution containing $Mg(NO_3)_2 \cdot 6H_2O$ (Sigma-Aldrich, Louis, MO, USA, 99% purity for analysis), $Al(NO_3)_3 \cdot 9H_2O$ (Sigma-Aldrich, Louis, MO, USA, 99% purity for analysis), and urea (Sigma-Aldrich, Louis, MO, USA, 99% purity for analysis) was prepared and complexed, at 50 °C for 1 h, under continuous stirring. Mass ratio of $Mg^{2+}$ to $Al^{3+}$ ions in solution was 0.65:0.35, and later obtained Mg–Al ratio in the oxide matrix was comparable to that of hydrotalcite-based samples reported elsewhere. The amount of urea was stoichiometric, in regard to the combustion reaction toward $Mg(NO_3)_2$ and $Al(NO_3)_3$. The purified foam was placed in the solution containing precursors and soaked for 1–2 min, while being constantly rotated or gently moved to allow penetration of the internal pores. Subsequently, the foam was placed in the ceramic boat and placed inside the furnace preheated to 500 °C for 10 min. The studied foam was cooled down. An increase in its mass was recorded with the laboratory balance after each soaking–heating cycle and repeated several times until the desired amount of Mg–Al layer was deposited. Nickel oxide was deposited using the same approach on the calcined foams coated with Mg-Al mixed oxides.

All the samples were reduced in the quartz fixed-bed reactor at 500 °C for 1 h. The heating rate for the reduction step was 10 °C/min under gas flow of 100 mL/min, composed of 5% $H_2$ in Ar.

The synthesized samples are listed in Table 4, with the corresponding amounts of the deposited support phase (Mg–Al mixed oxide) and active phase (NiO). The name of each sample should be read as follows: first two digits represent Ni concentration, middle three digits correspond to the amount of Mg–Al layer, and the last two digits inform about the foam porosity. The tabularized nickel content is assigned to the expected amount of the metallic nickel in the catalyst, after the reduction step, taking into account the mass of the entire foam. Two concentrations of nickel were applied, which were expected to be ca. 10 wt% and ca. 15 wt%, but eventually resulted in ca. 9 wt% and 12–13 wt%. Furthermore, two types of Mg–Al layer thicknesses were compared, ca. 100 mg and ca. 150 mg.

**Table 4.** List of the OCF samples and amounts of support phase and active phase deposited on the foams.

| Catalyst | Nickel Content (%) | NiO (mg) | Mg-Al (mg) | Volumetric Catalyst Loading g/l$_{foam}$ | Porosity (ppi) |
|---|---|---|---|---|---|
| 15_150_30 | 12.87 | 317 | 157 | 212.9 | |
| 10_150_30 | 8.71 | 230 | 155 | 176.0 | 30 |
| 15_100_30 | 12.61 | 314 | 120 | 192.2 | |
| 15_150_45 | 12.39 | 316 | 170 | 219.3 | |
| 10_150_45 | 9.60 | 239 | 157 | 180.7 | 45 |
| 15_100_45 | 13.26 | 339 | 114 | 199.4 | |

### 4.2. Physicochemical Characterization

The catalysts were characterized using the following methods: XRD, $H_2$-TPR, low-temperature $N_2$ sorption, $CO_2$-TPD, and SEM. X-ray diffraction (XRD) measurements were carried out with a Panalytical Empyrean diffractometer (MALVERN PANALYTICAL, Warsaw, Poland) for the bare open-cell foams, as-synthesized materials with the deposited coating, and reduced and post-run catalysts. The diffractometer working in Bragg–Brentano θ-θ geometry was equipped with Cu Kα (λ = 1.5406 Å) radiation. The recorded data were within a 2θ range of 10–90 deg. The average diameter of the Ni crystallite was calculated by Scherrer's equation, including the instrument broadening correction and the shape factor of 0.89 [55].

Low-temperature $N_2$ sorption measurements were performed with the Belsorp Mini II (BEL Japan, Osaka, Japan). Catalysts were outgassed for 2 h at 300° beforehand. Temperature-programmed reduction ($H_2$-TPR) and $CO_2$ temperature-programmed desorption ($CO_2$-TPD) measurements were performed with the BELCAT-M apparatus (BEL Japan, Osaka, Japan), equipped with a TCD detector. The catalysts were outgassed for 2 h at 100 °C before the $H_2$-TPR measurement and further reduced at the heating ramp of 10 °C/min, in the temperature range from 100 to 950 °C, under a flow of a gas mixture containing 5% $H_2$ in Ar (50 mL/min). $CO_2$-TPD was recorded for the reduced samples, directly after $H_2$-TPR measurement, with purging in He beforehand. A mixture of 10%$CO_2$/He was fed to adsorb $CO_2$ on the catalyst for 1 h. Then, the reactor was flushed with pure He (50 mL/min) for 15 min to remove weakly adsorbed carbon dioxide from the surface of the catalyst. $CO_2$-TPD measurements were carried from 100 to 600 °C with a heating rate of 10 °C/min. $CO_2$ uptake was calculated as the number of desorbed volumes of the gas, from the area under the experimentally obtained curve. The setup was calibrated prior to the measurement with a known amount of $CO_2$, to determine the precise area of the one pulse, as registered by the TCD detector (BEL Japan, Osaka, Japan).

Scanning electron microscopy (SEM) analyses were performed for bare open-cell foams and the reduced OCF-based catalysts with Hitachi SU-70 (Tokio, Japan)combined with an EDS (Energy-dispersive X-ray spectroscopy) detector electron microscope operating at 1–5 kV(Hitachi, Tokio, Japan).

### 4.3. Catalytic Tests

The studied catalysts were reduced for 1 h at 500 °C under the flow of 5%$H_2$/Ar (100 mL/min) prior to the test. Catalytic tests toward $CO_2$ methanation reaction were carried out inside the fixed bed tubular borosilicate glass U-type reactor (home made reactor, Sorbonne université, Paris, France) heated with the vertical furnace (Figure S2). To provide the maximal fitting of the foam-based catalysts, the specially designed reactors were prepared, in which foam was placed and stabilized during the process of glass blowing, to reduce the free space between the wall of the foam and the internal walls of the reactor. The temperature was registered with a K-type thermocouple directly placed on the external wall of the reactor at the catalytic bed. The reactor feed gases contained $CO_2$/$H_2$/Ar in the ratio of 1.5:6:2.5 with a total flow of 100 mL/min. GHSV, considering the volume of the foam placed inside the reactor, was 2 000 h$^{-1}$. The products of the reaction, as well as the unconverted reactants ($CO_2$, CO, $CH_4$, and $H_2$), were analyzed with an online micro-chromatograph (Agilent Varian GC4900, Agilent, Les Ulis, France) equipped with a thermal conductivity detector (TCD). Temperature-programmed surface reaction (TPSR) catalytic tests were performed in the temperature range from 250 °C to 450 °C. The sample was kept in steady-state operation for 30 min at each of the examined temperatures, with a heating rate between steps of 10 °C/min.

Equilibrium $CO_2$ conversion and selectivity to $CH_4$ were calculated with HSC Chemistry 5.0 software (METSO, Espoo, Finland) and are available in the Supplementary Materials as Table S1. The conditions assumed for calculations included pressure of 1 bar, $CO_2$/$H_2$ ratio in feed of 1:4, and temperature range from 100 to 500 °C.

The conversion of $CO_2$ conversion and the selectivity to $CH_4$ were calculated based on Equations (2) and (3).

$$CO_2 \text{ conversion } \chi_{CO_2} = \frac{F_{CO_2\text{inlet}} - F_{CO_2\text{outlet}}}{F_{CO_2\text{inlet}}} \tag{2}$$

$$CH_4 \text{ selectivity } \chi_{CH_4} = \frac{F_{CH_4\text{outlet}}}{F_{CO\text{outlet}} + F_{CH_4\text{outlet}}} \tag{3}$$

where F is the flow calculated from the concentration of the gases.

## 5. Conclusions

The structured open-cell foam-based catalyst for $CO_2$ methanation is evidently an interesting alternative to the pulverized one; however, the presented system does require further optimization for a possible scale-up process. The physicochemical properties of the synthesized catalysts are clearly important; however, the dominating factor affecting the catalytic performance at 300 °C, where the difference in the catalytic conversion was explicit, was the geometrical dimensions of the applied open-cell foams. OCFs with porosity of 45 ppi were more effective, considering both the activity in $CO_2$ hydrogenation reaction and the parameters resulting from it, describing the external diffusion under the studied test conditions. OCFs with porosity of 30 ppi showed lower activity at 300 °C and were associated with external mass transfer limitations due to the excessive pore size. Additionally, as applied in this study, the length of the microreactor was more than sufficient to provide the optimal contact time between the reactants and the surface of the catalyst and could be reduced to ca. 10–15 mm to still result in a Da-I number above 1, within its effective range.

**Supplementary Materials:** The following supporting information can be downloaded at https://www.mdpi.com/article/10.3390/catal14010011/s1. Figure S1: XRD diffractogram for $ZrO_2$ open-cell foam; Figure S2: Overview of the reactor design; Figure S3: Geometric parameters and pore diameter distribution of the open cell foams; Table S1: Equilibrium data for $CO_2$ methanation; Table S2: $CO_2$ conversion, $CH_4$ selectivity and standard deviation; Table S3: Geometrical characteristics of the studied open cell foams; Section S1: Supplementary characterization data; Section S2: Internal and external mass transfer limitations estimation methodology [41,62,63].

**Author Contributions:** Conceptualization, P.D.C. and P.S.; methodology, P.D.C. and P.S.; validation, P.D.C.; formal analysis, P.D.C. and P.S.; investigation, P.D.C. and P.S.; resources, P.D.C. and M.M.; data curation, P.D.C.; writing—original draft preparation P.S.; writing—review and editing, P.D.C.; visualization, P.D.C.; supervision, P.D.C. and M.M.; project administration, P.D.C.; funding acquisition, P.D.C. All authors have read and agreed to the published version of the manuscript.

**Funding:** This research received by French Embassy in Poland for providing the BGF Doctorate Cotutelle scholarship.

**Data Availability Statement:** Data are contained within the article and supplementary materials.

**Acknowledgments:** Thanks are due to Lanik s.r.o. for providing the raw supports.

**Conflicts of Interest:** The authors declare no conflicts of interest.

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
