# Peer review of "Optimization of an Open-Cell Foam-Based Ni-Mg-Al Catalyst for Enhanced CO2 Hydrogenation to Methane"

_catalysts, doi:10.3390/catal14010011_

Round 1
Reviewer 1 Report
Comments and Suggestions for Authors
In this work, the performance of a structured nickel catalyst, viz., nickel supported on Mg-Al oxide-coated ZrO2 open-cell foam (OCF), in the hydrogenation of CO2 to methane was investigated. The results indicate that the catalyst with 45 ppi foam having denser pores displays higher activity in CO2 methanation, owing to less radial transport limitation, compared with that obtained with 30 ppi foam having sparser pores.
The application of structured catalyst in the hydrogenation of CO2 is meaningful and the related research work is really interesting. However, it seems to this reviewer that current manuscript is rather rough and immature for publication in the journal of Catalysts, due to following issues.
(1) The actual active component and its intrinsic activity in the hydrogenation of CO2 to methane were not properly determined. The active component should be the dispersed Ni species interacted with the Mg-Al oxide coating layer. The intrinsic catalytic activity should be related to both the number of the active Ni species and the activity of the Ni species depending on the Ni–Mg-Al interaction.
(2) The effect of mass transfer limitation on the CO2 methanation may not be properly considered. Such a consideration should in general be conducted by varying the coating thickness as well as the space velocity. The test data are actually insufficient to get any deep insight on the limitation between kinetic reaction and mass transfer. In general, compared with the reaction limitation, the mass transfer limitation becomes more serious at a higher temperature, which differs from that observed in this work.
(3) Current manuscript is rather rough and has many confusing statements, like:
(a) Line 34, ΔH0298 = -165 kJ, wrong unit;
(b) Line 197, “Therefore, the specific surface area of the bare foam is assumed to be less than 1 cm3/g”?
(c) Line 270, “from medium-strenght sites”?
(d) Line 376, “an increment in the thickness of the deposited coating may result in the internal mass transfer rate higher than that of the studied hydrogenation reaction”?
(4) Other points:
(a) Fig. 4, the crystal size of Ni particulates may also be provided, like Fig. 10;
(b) Fig. 6, the scale bar is unclear;
(c) Fig. 8, the coordinate scale is improper and suitable guide lines may be added for the experimental points;
(d) Line 392, “…presented in Figure 6 and 7? (e) Line 402, “Open-cell foams structure is consisted of cell units…”;
(f) Line 405, “The kinetics on which calculations were based was assumed for the first order reaction”?
(5) Detailed conditions for calculating the reaction equilibrium of CO2 methanation, like pressure, initial H2/CO2 feed, and so on, should be provided (Fig. S1).
Comments on the Quality of English Language
Extensive editing of English language is required.
Author Response
In this work, the performance of a structured nickel catalyst, viz., nickel supported on Mg-Al oxide-coated ZrO2 open-cell foam (OCF), in the hydrogenation of CO2 to methane was investigated. The results indicate that the catalyst with 45 ppi foam having denser pores displays higher activity in CO2 methanation, owing to less radial transport limitation, compared with that obtained with 30 ppi foam having sparser pores.
The application of structured catalyst in the hydrogenation of CO2 is meaningful and the related research work is really interesting. However, it seems to this reviewer that current manuscript is rather rough and immature for publication in the journal of Catalysts, due to following issues.
- The actual active component and its intrinsic activity in the hydrogenation of CO2 to methane were not properly determined. The active component should be the dispersed Ni species interacted with the Mg-Al oxide coating layer. The intrinsic catalytic activity should be related to both the number of the active Ni species and the activity of the Ni species depending on the Ni–Mg-Al interaction.
Answer: Thank you for this remark. The catalytic activity of similar Ni-Mg-Al catalytic systems is dependent on concentration of nickel and also number of available surface basic sites [Wierzbicki et al., The influence of nickel content on the performance of hydrotalcite-derived catalysts in CO2 methanation reaction, International Journal of Hydrogen Energy, Volume 42, Issue 37, 2017, p. 23548-23555; Wierzbicki et al.; Novel Ni-La-hydrotalcite derived catalysts for CO2 methanation, Catalysis Communications, Volume 83, 2016,Pages 5-8]. In that matter, both sets of tested samples were comparable between each other. The only significantly differing factor between the studied systems was geometry of foams and based on this presented conclusion were drawn.
- The effect of mass transfer limitation on the CO2 methanation may not be properly considered. Such a consideration should in general be conducted by varying the coating thickness as well as the space velocity. The test data are actually insufficient to get any deep insight on the limitation between kinetic reaction and mass transfer. In general, compared with the reaction limitation, the mass transfer limitation becomes more serious at a higher temperature, which differs from that observed in this work.
Answer: Thank you for this remark. Due to the setup limitation, authors focused on varying the coating thickness and cell dimensions of the studied foams, however, the obtained dataset strongly suggests that under the applied reaction conditions the external mass transfer limitation effect is visible at 300 oC. At higher reaction temperatures, above 350 oC indeed is expected to be higher, but due to the high activity of the catalyst, associated also at this temperature regime to improved kinetics of the reaction, the mass transfer limitations were not explicit.
(3) Current manuscript is rather rough and has many confusing statements, like:
(a) Line 34, ΔH0298 = -165 kJ, wrong unit;
(b) Line 197, “Therefore, the specific surface area of the bare foam is assumed to be less than 1 cm3/g”?
(c) Line 270, “from medium-strenght sites”?
(d) Line 376, “an increment in the thickness of the deposited coating may result in the internal mass transfer rate higher than that of the studied hydrogenation reaction”?
Answer: Thank you, we corrected those points.
(4) Other points:
(a) Fig. 4, the crystal size of Ni particulates may also be provided, like Fig. 10;
Answer: The Ni° crystal size is provided in the Table 3 p 11.
(b) Fig. 6, the scale bar is unclear;
Answer: Thank you, we corrected the Figure.
(c) Fig. 8, the coordinate scale is improper and suitable guide lines may be added for the experimental points;
Answer: We kept this scale from zero to 100 % for conversion and from 250 to 450°C for temperature, since the catalysts are inactive at 250°C.
(d) Line 392, “…presented in Figure 6 and 7? (e) Line 402, “Open-cell foams structure is consisted of cell units…”;
Answer: We corrected the text.
(f) Line 405, “The kinetics on which calculations were based was assumed for the first order reaction”?
Answer: We corrected the text.
(5) Detailed conditions for calculating the reaction equilibrium of CO2 methanation, like pressure, initial H2/CO2 feed, and so on, should be provided (Fig. S1).
Answer: The conditions were added to the text.

Reviewer 2 Report
Comments and Suggestions for Authors
The authors present a paper showing the synthesis, characterisation and catalytic testing of Ni deposited on an open cell foam. The catalysts are characterised with a range of techniques (PXRD, BET, TPR, SEM) which provide a robust analysis of the materials and helps to define their structure-activity relationships. Catalytic activity appears to be modest but a strong preference for the formation of CH4 (over CO) is identified. I think this is a robust scientific investigation and is suitable for publication in Catalysts. I have a few points for the authors to address:
The PXRD data is complex due to the number of species present. This is unavoidable. However I do think they could be presented in a clearer way. Do we need all three spectra for 30 and 45ppi? Might these be better in the SI and one of each along with the ZrO foam be better in Fig 1? It is also difficult to compare these spectra to those later on.
Is it possible that the TPR measurements are detecting a reduction of Zr(IV) as well?
The scales on the SEM measurements are not visible. These could be included in the figure caption?
The reporting of the catalytic studies should be more detailed in section 2.2. What methods were used to detect the gases? What quantities of CH4 were formed? What was the path length of the reaction vessel? I noticed that the flow rate is high, perhaps this affects the conversion?
A number of the acronyms used in the paper are not defined. I'd urge the authors to address this as it affects the readability for non-experts in this field. (eg ppi/TPR/TDP/TPSR/etc which dont appear to be defined).
Some trivial points, I noticed a few typos (L270 'strength', L355 'suggesting', the incorrect symbol for degrees C is used in several places eg L219)
Author Response
The authors present a paper showing the synthesis, characterisation and catalytic testing of Ni deposited on an open cell foam. The catalysts are characterised with a range of techniques (PXRD, BET, TPR, SEM) which provide a robust analysis of the materials and helps to define their structure-activity relationships. Catalytic activity appears to be modest but a strong preference for the formation of CH4 (over CO) is identified. I think this is a robust scientific investigation and is suitable for publication in Catalysts. I have a few points for the authors to address:
The PXRD data is complex due to the number of species present. This is unavoidable. However I do think they could be presented in a clearer way. Do we need all three spectra for 30 and 45ppi? Might these be better in the SI and one of each along with the ZrO foam be better in Fig 1? It is also difficult to compare these spectra to those later on.
Is it possible that the TPR measurements are detecting a reduction of Zr(IV) as well?
Answer: Actually, till now there is no indication of a possible reduction of zirconium oxide during TPR neither in this study, neither in other studies conducted before. Authors performed H2-TPR on bare foams, but no reduction was detected. It might be associated to the very high stability of the industrial foams, used in general for operation at temperatures above 1000 oC.
The scales on the SEM measurements are not visible. These could be included in the figure W
Answer: Thanks, we corrected the plot.
The reporting of the catalytic studies should be more detailed in section 2.2. What methods were used to detect the gases? What quantities of CH4 were formed? What was the path length of the reaction vessel? I noticed that the flow rate is high, perhaps this affects the conversion?
Answer: The methods are pointed out. We used a Micro-CG equipped with TCD detector. Methane formation varied from 0 to 27% of the outlet gas stream giving in the case of maximal conversion ca. 0.2 mol CH4/h.
The selectivity in methane is about 98%, see Line 324
The reactor length was 30 cm based on a U shape reactor
The inner diameter in which catalyst was placed was about 9.5 mm.
We kept large flow rate in order to avoid extra diffusion.
A number of the acronyms used in the paper are not defined. I'd urge the authors to address this as it affects the readability for non-experts in this field. (eg ppi/TPR/TDP/TPSR/etc which dont appear to be defined).
Answer: We corrected the text
Some trivial points, I noticed a few typos (L270 'strength', L355 'suggesting', the incorrect symbol for degrees C is used in several places eg L219)
Answer: We corrected the text
Reviewer 3 Report
Comments and Suggestions for Authors
In my opinion, the research presented in the manuscript was carried out at a good experimental level. The authors use a variety of methods for studying catalysts and mathematical calculations.
There are several notes and comments.
1. Are the authors sure that they do not have exothermic heating of the catalysts? After all, the thermocouple measures the temperature of the reactor wall, and not the catalyst volume.
2. How the reduced samples were transferred for testing and how the same was done for samples after catalytic tests.
3. Conclusions to table 3. Can nickel block strong basic centers?
4. The authors compare catalysts in terms of co2 conversion at temperatures of 250 and 300°C. However, at a temperature of 250, the catalysts are not active, and at a temperature of 300°C, 45 ppi foams have a conversion very close to equilibrium. Would it be better to compare 30 and 45 ppi foam catalysts at 275oC? In addition, from the catalytic test it is not clear which catalyst is better: 15_100_45, 10_150_45 or 15_150_45; their CO2 conversions at 300°C are the same. Maybe it is necessary to conduct tests at high flows of reagents or lower the temperature?
5. The authors believe that the low activity of the catalyst based on 30 ppi foam is associated with difficulty in mass transfer, maybe this is not true? The low activity may be due to the low foam surface of 30 ppi. When the active component is deposited, a thick layer of it is formed and not all of the active component participates in the reaction, especially since the data on the specific surface area confirm this.
There are several comments on the text.
1. It is better to remove Figure 2 in Supplementary Information.
2. In the title of Table 2 only one sample is indicated, although the data is for two samples.
3. How was Sbet (m2/gcoating) calculated?
4. There is an error in the words elsewhere, line 209. Look at the text for errors.
Author Response
In my opinion, the research presented in the manuscript was carried out at a good experimental level. The authors use a variety of methods for studying catalysts and mathematical calculations.
There are several notes and comments.
- Are the authors sure that they do not have exothermic heating of the catalysts? After all, the thermocouple measures the temperature of the reactor wall, and not the catalyst volume.
Answer: Exothermic heating of the catalyst is present. The temperature of the empty reactor was correlated with the temperature of the oven and compared to the temperature of the reactor with the active catalyst inside. Based on those calibrations the further adjustment of the oven temperature was made to obtain possibly realistic test conditions.
- How the reduced samples were transferred for testing and how the same was done for samples after catalytic tests.
Answer: The catalysts are reduced in-situ prior reaction there is no transfer of the catalysts from one test to another one. After run, the reactor is cutted and the catalyst placed in Glove box for further characterization.
- Conclusions to table 3. Can nickel block strong basic centers?
Answer: Actually, no the Ni° at this concentration, do not really affect the basicity itself.
[Wierzbicki et al., The influence of nickel content on the performance of hydrotalcite-derived catalysts in CO2 methanation reaction, International Journal of Hydrogen Energy, Volume 42, Issue 37, 2017, p. 23548-23555]
- The authors compare catalysts in terms of co2 conversion at temperatures of 250 and 300°C. However, at a temperature of 250, the catalysts are not active, and at a temperature of 300°C, 45 ppi foams have a conversion very close to equilibrium. Would it be better to compare 30 and 45 ppi foam catalysts at 275oC? In addition, from the catalytic test it is not clear which catalyst is better: 15_100_45, 10_150_45 or 15_150_45; their CO2 conversions at 300°C are the same. Maybe it is necessary to conduct tests at high flows of reagents or lower the temperature?
Answer: Thank you for this remark. Due to the special reactor design, authors are not able to repeat the test on the same sample at lower temperature. At lower temperature, the catalysts are inactive. At higher flow, the experiments were not performed since we did not want to change many parameters in this study to maintain the possibility of comparing the obtained results with the other analogous catalysts from which the study begun.
- The authors believe that the low activity of the catalyst based on 30 ppi foam is associated with difficulty in mass transfer, maybe this is not true? The low activity may be due to the low foam surface of 30 ppi. When the active component is deposited, a thick layer of it is formed and not all of the active component participates in the reaction, especially since the data on the specific surface area confirm this.
Answer: The thickness of the coating for samples based on 30 ppi foam is not significantly higher than comparing to the 45 ppi foam counterparts. Also, Da-III and Carberry number did not suggest strong internal diffusion limitations for the samples based on 30 ppi. Based on the results, the most significant factor varying the samples are the geometric dimensions of the foam.
There are several comments on the text.
- It is better to remove Figure 2 in Supplementary Information.
Answer: Thank you for this suggestion, authors would like to keep the figure having in mind less experienced researchers.
- In the title of Table 2 only one sample is indicated, although the data is for two samples.
Answer: We corrected the text.
- How was Sbet (m2/gcoating) calculated?
Answer: The value of SBET per gram of the catalyst war corrected using only the mass of deposited coating/mass of the entire sample, considering the mass of the foam as 1.
- There is an error in the words elsewhere, line 209. Look at the text for errors.
Answer: We corrected the text.

Round 2
Reviewer 1 Report
Comments and Suggestions for Authors
After re-read the revised manuscript, this reviewer feels that the major concerns, which hinder its acceptance for publication in Catalysts, remain as before.
The higher activity of the 15_150_45 catalyst compared with the 15_150_30 one is ascribed by the authors to the less external mass transfer limitation of the former. However, this is only based on the some estimated dimensionless numbers. This reviewer feels that the effect of mass transfer limitation on the CO2 methanation may not be properly considered. Such a consideration should in general be conducted by varying the coating thickness as well as the space velocity at different temperatures. The test data presented in current work are actually insufficient to get any deep insight on the limitation between kinetic reaction and mass transfer. In general, compared with the reaction limitation, the mass transfer limitation becomes more serious at a higher temperature, which differs from that observed in this work. On the contrary, 15_150_45 and 15_150_30 display rather great differences in the textural properties (e.g., surface area of 165 vs. 111 m2/g, Table 2) and surface basicity (44 vs. 20 μmol/g, Table 3) and perhaps other the dispersion and nature of active sites, which may contribute significantly to their difference in the catalytic activity.
In addition, the writing of this paper remains rather poor, like:
“Emission of greenhouse gases is currently one of the biggest concerns due to associated with it climate change and an increase in the global mean surface air temperature”?
“due to the kinetic limitations reactions is normally carried out at the temperature range between 250-450 °C”?
“cpsi (counts per scare inches)”?
“the specific surface area of the started material (ZrO2 foam) is assumed to be less than 1 cm3/g”?
Comments on the Quality of English Language
Extensive editing of English language required
Author Response
The higher activity of the 15_150_45 catalyst compared with the 15_150_30 one is ascribed by the authors to the less external mass transfer limitation of the former. However, this is only based on the some estimated dimensionless numbers. This reviewer feels that the effect of mass transfer limitation on the CO2 methanation may not be properly considered. Such a consideration should in general be conducted by varying the coating thickness as well as the space velocity at different temperatures. The test data presented in current work are actually insufficient to get any deep insight on the limitation between kinetic reaction and mass transfer. In general, compared with the reaction limitation, the mass transfer limitation becomes more serious at a higher temperature, which differs from that observed in this work. On the contrary, 15_150_45 and 15_150_30 display rather great differences in the textural properties (e.g., surface area of 165 vs. 111 m2/g, Table 2) and surface basicity (44 vs. 20 μmol/g, Table 3) and perhaps other the dispersion and nature of active sites, which may contribute significantly to their difference in the catalytic activity.
Answer: Thank you for your remarks. Authors improved the text and the explanation of the results according to your suggestions. The paragraph was added.
As it was reported elsewhere, for the Ni-Mg-Al mixed-oxide system that such physicochemical properties as specific surface area and nickel dispersion are less influential factors regarding the CO2 conversion of methanation catalyst, then i.e. surface basicity [11,12]. Considering the differences of basic sites distribution throughout the series of samples based on foams with 30 and 45 ppi, it is visible that despite the varying availability of surface basic centers associated to the different coating thickness, catalysts based on the same geometry of foam, show comparable catalytic activity. It was strongly pointed out in the literature, that the changes in the geometry of the structured catalyst, are visibly affecting the activity, associating it with the diffusion of gases through the catalytic bed, and eventual heat transfer [24–28]. Thus, the presented conclusions are implying that the geometric dimensions, which are the most significant differentiating factor, are affecting the catalytic activity of the structured catalyst.
In addition, the writing of this paper remains rather poor, like:
“Emission of greenhouse gases is currently one of the biggest concerns due to associated with it climate change and an increase in the global mean surface air temperature”?
“due to the kinetic limitations reactions is normally carried out at the temperature range between 250-450 °C”?
“cpsi (counts per scare inches)”?
“the specific surface area of the started material (ZrO2 foam) is assumed to be less than 1 cm3/g”?
Answer: Thank you for pointing out the errors in the vocabulary, authors revised the text with a native speaker.